# In Situ Rapid Hierarchical Growth of Ag Nanodendrites in *Phytantriol*: Influence on Polydispersity and Optical Characteristics

**DOI:** 10.3390/nano14181534

**Published:** 2024-09-22

**Authors:** Nisrin Alnaim

**Affiliations:** Department of Physics, College of Science, King Faisal University, Al-Ahsa, P.O. Box 400, Hofuf 31982, Saudi Arabia; nalnaim@kfu.edu.sa

**Keywords:** phytantriol, silver dendrites, room temperature growth, structural analysis

## Abstract

Silver nanodendrites (AgNDs) were effectively synthesized utilizing a phytantriol template at ambient temperature without electrodeposition. In comparison to AgNDs, the phytantriol-templated silver nanodendrites (P/AgNDs) exhibited a smoother structure with the well-ordered growth of smaller particles around 51 nm. Moreover, the P/AgNDs exhibited uniform elemental distribution, forming chemical bonds with functional groups, crystallite size around 42 nm, and high transmittance around 95%. These experimental findings indicate that room temperature-based hierarchical P/AgNDs have considerable potential for diverse applications, specifically in sensing.

## 1. Introduction

Nanoscale dimension-based hierarchical structures can provide excellent opportunities to improve the performance of materials for a variety of technological applications [1]. Such hierarchies’ nanostructures are made up of neatly arranged nanoscale building blocks with 0D, 1D, 2D, or 3D architectures. These could be a combination of nanostructures like quantum dots, nanospheres, nanowires, nanorods, nanofibers, nanoflowers, and nanodendrites [2,3]. Despite the fact that metallic dendrites were traditionally produced on electrode surfaces via electrochemical deposition [4], these metallic nanodendrites, a higher-order form of 1D nanomaterials with a hierarchical structure, hold potential for applications in plasmonics and biosensing. Due to their distinct optical and electrical properties, one-dimensional (1D) Ag nanostructures have been widely studied. The hierarchically structured Ag dendritic fractals, commonly observed in non-equilibrium growth processes, have garnered significant interest in recent times. These formations serve as an inherent model for exploring disordered systems theoretically. These dendritic nanomaterials exhibit considerable potential in various applications such as surface-enhanced materials, catalysts, sensors [5], and optical and electronic systems, owing to their intricate fractal hierarchical structure [6]. Ag nanodendrites (AgNDs) comprise primary trunks and numerous hierarchical branches adorned with attached leaves. These formations offer an extensive surface area and intricate interfacial structures, significantly augmenting specific desired characteristics [6]. AgNDs could be synthesized using the sol–gel method, chemical bath deposition, electrodeposition [7], and template-assisted processes [8], including mesoporous silica and tubular protein assemblies. Additionally, a nanoparticle-to-nanowire formation mechanism has been reported for synthesizing Ag nanowires in supercritical water. Chen et al. [9] reported AgNDs using poly(vinyl alcohol) through photoreduction, while Xie et al. [10] used Reney nickel as a template and reducing agent for AgNDs. Moreover, among template-assisted processes, a soft template can be used to synthesize hierarchical nanostructures/nanodendrites. A soft template, such as phytantriol, has been used to synthesize 1D and 3D nanostructures [11,12]. Phytantriol is an amphiphilic molecule that can self-assemble to form a range of periodic nanostructures (mesophases) when they are combined with water. A diamond inverse bicontinuous cubic phase ($Q_{II}^{D}$) mesophase has been extensively used as a template due to its stability in excess water and ability to form at room temperature [11,12]. Furthermore, AgNPs aggregation-based detection is a common approach due to their sensitivity to changes in shape, size, or environment, which affect their SPR properties. AgNPs with surface plasmon resonance (SPR) bands can function through aggregation in various applications, including Surface-Enhanced Raman Scattering and sensing. Many traditional AgNP-based sensors suffer from instability over time due to uncontrolled aggregation or oxidation. The phytantriol template may increase the stability of AgNPs in different environments, reducing their tendency to degrade or aggregate nonspecifically, thus prolonging their shelf-life and maintaining consistent detection performance. The phytantriol template could modulate the surface chemistry, making the system more responsive to specific analytes. Therefore, the aggregation of P/AgNDs is preferred over that of AgNPs due to several points, such as controlled morphology, enhanced stability, and surface functionalization. Often, the electrodeposition process is used to synthesize nanostructure materials through a soft template [13,14]. In the electrodeposition process, a deposition voltage for a target metal through the template must be known, which varies from one metal to another. The optimum deposition could be found by conducting a series of electrochemical fabrications for nanostructure materials at different ranges of deposition voltages. This process might consume time and effort. However, an oxidation–reduction reaction is one of the most popular chemical methods that can be used because of its relative simplicity and affordable costs. Therefore, in the current study, an oxidation–reduction reaction with a phytantriol template was used to synthesize AgND hierarchies on Cu wire. In contrast to the mentioned reports, the reaction procedure does not require any supplementary reagents like reductants or surfactants or involve stringent conditions such as high temperatures or light exposure. Moreover, the advantages of this method include its heterogeneous nature, amenability to control, and high yield. The P/AgNDs exhibited uniform elemental distribution and particle size of 51 nm, heterogeneous bonding small crystallite size, and high transmittance. Further, it is significant to mention here that the synthesis of AgNDs at room temperature with a phytantriol template has never been reported.

## 2. Experimental Section

### 2.1. Method

The synthesis steps of AgNDs are shown in Figure 1. First, to control the growth through phytantriol (3,7,11,15-tetramethyl-1,2,3-hexadecane-triol), one end of the Cu wire was covered by epoxy Resin, from Graffiti Resin Company, Illinois, USA, while the other end was submerged into a 2:1 mixture of phytantriol/ethanol. The Cu wire was left at room temperature for 30 min to evaporate the ethanol to obtain a thin film of phytantriol on the Cu wire. Then, the wire was submerged in DI water for about 2 h to form $Q_{II}^{D}$ an inverse bicontinuous cubic phase of a phytantriol template. After that, the wire was submerged into 0.1 M AgNO_3_ for Ag particles to pass through the phytantriol water channels and fill them. The balanced reaction between phytantriol and AgNO_3_ on Cu wire is as follows
C_20_H_42_O_3_ + 3 AgNO_3_ + Cu →3 Ag + C_20_H_41_O_3_ + HNO_3_+ Cu (NO_3_)_2_

### 2.2. Characterizations

The morphology of synthesized nanodendrites was revealed by the Tescan scanning electron microscope (SEM) (Tescan vega 3 SBU, Brno, Czech Republic). Samples were mounted on aluminum microscopy stubs using carbon tape and then coated with gold (Au) for 120 s using the Quorum Techniques Ltd. sputter coater (Q150t, Laughton, UK). The nanoparticles were calculated by the ImageJ software (V1.54k). Elements with an atomic weight (%) were determined by the EDX (Energy Dispersive spectrometer) model, Bruker Nano GmbH Berlin, Germany. The Esprit 2.0. Fourier transform infrared spectrometer (FTIR) Agilent Cary 630 (Santa Clara, California, USA) was used to evaluate the intermolecular interaction between the phytantriol and AgNDs in the range of 650–4000 cm^−1^ with a resolution of 2 cm^−1^. The X-ray diffractometer (XRD) model Rigaku Ultima IV(Tokyo, Japan) was used to determine the crystallinity of the nanodendrites. Transmission in the spectral range of 400–700 nm was measured by a Shimadzu UV-3101PC spectrophotometer (Kyoto, Japan).

## 3. Results and Discussion

In AgNDs and P/AgNDs, the oxidation and reduction states of Ag primarily pertain to the transition between Ag^+^ (oxidized state) and Ag^0^ (reduced state). After the reduction process, Ag^0^ forms the dendritic structure while Cu facilitates this reduction. The Ag^0^ atoms aggregate and grow into dendrites. The stability and interaction of Ag^0^ with Cu and phytantriol may be influenced by the formation of weak interactions or surface passivation, which helps to stabilize the dendritic structure and prevent further oxidation back to Ag^+^. This ensures that the silver remains in its reduced state, crucial for applications such as plasmonic or sensing devices. According to the literature [15], state-free elements always have an oxidation state of zero and mostly +1 for compounds.

Furthermore, when the Cu wire was dipped into the AgNO_3_ solution, the Ag species reduced at the Cu wire surface and formed the Ag nanoparticles (AgNPs) immediately. As time progressed, the AgNPs started to self-assemble into the AgNDs. The AgNDs in Figure 2a show a thickness of around 13 mm for 10 min, while P/AgNDs’ thickness was estimated to be at around 1.34 mm [Figure 2b]. Further, P/AgNDs’ growth was estimated to be around 2.15 mm after 60 min, as shown in Figure 2c, probably due to Ag species traveling through the phytantriol template with water channels (radius of around 2.34 nm), which was resisting the flow of Ag species, in agreement with the other researchers [16]. The AgNDs without the phytantriol template possessed cluster structures with sharp needles, as shown in Figure 2d, while small branches emerging from the main trunk were observed with the phytantriol template [Figure 2e,f]. The hierarchical dendritic structure can be attributed to the kinetic reaction, which depends on the chemical reaction rates and diffusion. According to the diffusion-limited aggregation (DLA) theory, the small branches (dendrites) grow out due to the diffusive fluxes more swiftly than on a flat surface, as documented in the literature [17,18]. Furthermore, the morphology of AgNDs was influenced by a balance between kinetic factors and thermodynamics. Initially, during short reaction times, kinetic factors predominate, driven by the concentration gradient of Ag ions, which governs the rapid nucleation and early-stage growth of the dendrite structure. As the reaction proceeds and time increases, thermodynamic factors begin to take over, leading to the relaxation of small grains and the formation of the hierarchical structure of the dendrite.

This growth mechanism, where a dendrite head forms at the final stage, was attributed to the oriented attachment model. In this process, nanoparticles align along specific crystallographic directions, leading to their aggregation and coalescence. Oriented attachment facilitates the fusion of smaller particles with the growing dendrite tip, thereby accelerating dendrite elongation. As the nanoparticles merge through this attachment mechanism, they contribute to the further growth and branching of the AgNDs, aligned with the literature [19].

The SEM images of AgNDs and P/AgNDs are shown in Figure 3a,b, respectively. Figure 3a reveals the hierarchical random arrangement of AgNDs. These nanodendrites were composed of nanoparticles, which are illustrated in a magnified view of AgNDs [Figure 3b]. The nanoparticle distribution was estimated with an average diameter of 140 nm [Figure 3c], whereas, after the phytantriol template [Figure 3d], a uniform arrangement of AgNDs was observed with the spherical-shaped nanoparticles linearly arranged in an ordered manner, as shown in Figure 3e. The P/AgNDs revealed nanoparticles with an average diameter of 51 nm, as illustrated in Figure 3f. The dendritic structure comprises particles of varying sizes. Consequently, the Gaussian particle distribution curve was used to represent the average nanoparticle size. These results indicate that a suitable concentration gradient of Ag nanocrystals was essential for the successful growth of well-defined nanodendrites in a static solution. Moreover, the size difference can be attributed to kinetic factors associated with the nucleation and growth of the nanoparticles, corresponding to the literature [20]. As the Ag species settle into the nanoscale water channels, the phytantriol template regulates nucleation, slows down growth kinetics, and prevents the rapid growth or agglomeration of nanoparticles, resulting in smaller particles. Sun et al.’s [21] research aligns with this observation, demonstrating that polymer scaffolding plays a key role in controlling the morphology and size of nanoparticles. In the absence of such scaffolding, these controls are lost, leading to faster growth kinetics and fewer constraints, causing the nanoparticles to grow larger. Furthermore, P/AgNDs underwent further aggregation and coalescence, forming 20–40 nm particles/crystalline nuclei, as depicted in Figure 3f. These nuclei began to exhibit a more ordered structure. The aggregation of these crystallites prompted reorientation, ultimately leading to the formation of dendrites. These dendrites, which ranged from 50 to 100 nm in size [Figure 3f], represented a more defined and organized phase in the growth of the nanodendrites. The reorientation and agglomeration of nanoparticles/crystallites were key in shaping these hierarchical dendritic structures, as known from the literature [19].

Figure 4a–d demonstrate the EDX mapping and spectra of AgNDs and P/AgNDs. Figure 4a,b show the uniform distribution of elements throughout the surface. The uniform distribution of elements suggests effective coverage and stabilization, which prevents aggregation and enhances the stability of the material. In both samples, Ag, N, and O signals along with C were observed, which belong to AgNO_3_ and phytantriol (C_20_H_42_O_3_), Figure 4c,d. The Cu peaks in both samples belong to the wire/substrate, whereas the carbon (C) presence in AgNDs might be from the air. The atomic weight (%) variations in the elements after the phytantriol template indicate that the phytantriol agents were indeed interacting with the AgNO_3_ molecules. The Wt. (%) of AgNDs and P/AgNDs was tabulated in the inset of Figure 4c,d. The P/AgNDs exhibited more uniform and controlled morphologies due to the stabilizing effect of phytantriol. This stability can prevent nonspecific aggregation, leading to more reliable and reproducible detection results in sensing.

The AgND and P/AgND bands are shown in Figure 5a,b, respectively. The band at 1044 cm^−1^ was due to the stretching of C–O from phytantriol, while a little intense band at 1092 cm^−1^ corresponds to the CO_3_Ag^−l^, as observed by the other researchers [22]. In P/AgNDs, a small feature at 874 cm^−1^ was due to the distorted vibration in the functional group NH_2_, in correspondence with the literature [23]. A little intense feature at 1361 cm^−1^ was assigned to the symmetry stretching of N=O from AgNO_3_ [24]; this band presence in AgNDs indicated interactions of AgNPs in the phytantriol [25]. The band at 1636 cm^−1^ corresponds to the stretching of C–N, suggesting the interaction of phytantriol with AgNPs [23]. The features at 2906 cm^−1^ and 2987 cm^−1^ were ascribed to the aliphatic C-H stretching vibration of CH_3_ and N-H bending vibration from phytantriol-templated AgND species [17,18]. The broadband at 3293 cm^−1^ was attributed to AgOH^-^, which helps to prevent aggregation and enhances the long-term stability of the nanoparticles, as documented in the literature [5]. The spectra suggested that the AgND growth kinetics were ruled by the phytantriol carbonyl group, and the oxidation of the -OH groups resulted in the formation of spheroidal AgNPs. However, dendrite-like structures were dominant and composed of spherical nanoparticles. Therefore, it can be inferred that the phytantriol molecules in P/AgNDs possessed functional groups that can enhance the interaction with specific targets, allowing for a more tailored sensor design. This could lead to the development of more precise and targeted detection mechanisms, beyond simple aggregation, which may increase their effectiveness in complex environments.

The XRD pattern of AgNDs and P/AgNDs is shown in Figure 6a,b. The diffraction peaks are located at 38.2°, 44.4°, 64.5°, and 77.4°, corresponding to the (111), (200), (220), and (311) planes of the face-centered cubic (FCC) Ag crystals according to the JCPDS No: 65 2871, in agreement with other researchers [26,27]. The crystallite size was estimated for the plane (111) by Scherrer’s equation
D = Kλ/βCosθ
where K signifies the crystallite shape factor, typically taken as 0.9, λ = 1.5406 represents the wavelength of the incident X-rays, β represents the full width at half maximum (FWHM) of the respective XRD peak, and θ corresponds to the diffraction angle. The crystallite size around 73 nm was calculated for AgNDs, which was reduced to 42 nm after the phytantriol template, indicating a good interaction of phytantriol with AgNDs.

Figure 7a,b show the transmittance spectra of AgNDs and P/AgNPs. No absorption band was observed within the visible range of both samples, suggesting no aggregation in self-assembled AgNDs before and after the phytantriol support. Transmittance of around 87% was observed for AgNDs at 550 nm, which increased up to 95% after phytantriol; this might be due to the porous structure. According to the literature [28], the size of the pores governs the transmittance; open and large pore-based structures enhanced the optical transparency of the material and permeability directed by Rayleigh scattering. Furthermore, the shape and size distribution of the nanoparticles also influenced the absorption and transmittance of the material due to the interaction of light with the nanoparticles. Oh et al. [29] reported that subwavelength-sized particles can form a homogeneous layer with an effective refractive index between that of Ag and air. This layer could smooth the transition in the refractive index from air to the substrate, thereby reducing surface reflectance and consequently increasing transmittance. Therefore, P/AgNDs exhibited high transmittance due to their small size compared to AgNDs.

The absorption spectra of AgNDs were largely governed by their plasmonic characteristics, which are influenced by the shape, size, and morphology of the dendritic structures. For AgNDs, the absorption peak position can vary based on the size and density of the dendrite arms. Smaller branches tend to shift the LSPR peaks toward the blue region, while larger branches shift the peaks toward the red region. Aggregation can also amplify the absorption intensity due to enhanced surface area and plasmon coupling. Figure 8a,b show the abortion spectra of AgNDs and P/AgNDs, respectively. Both samples illustrated a strong SPR band with good symmetry at wavelengths of 301 nm and 303 nm, suggesting small, spherical, and narrow size distribution interval-based nanoparticles. Mukherji et al. [30] reported Ag nano prisms by the absorption peak at 397 nm. Agnihotri et al. [31] reported SPR peaks at 393 nm, 394 nm, and 398 nm, corresponding to AgNPs of average sizes of 5 nm, 7 nm, and 10 nm, respectively. Furthermore, a change in the refractive index around AgNDs can shift to the peak position, making them useful in sensing.

## 4. Conclusions

A simple and quick method was used for preparing hierarchical AgNDs with the support of the phytantriol template at room temperature. The P/AgNDs revealed a smooth structure with the ordered growth of small-sized particles at 51 nm compared to AgNDs at approximately 140 nm. The P/AgNDs exhibited uniform elemental distribution, chemical bonding with functional groups, crystallite size around 42 nm, and high transmittance around 95%. Therefore, it can be concluded that due to improved stability, controlled aggregation, surface functionality, and high transmittance, P/AgNDs have potential for multimodal sensing, making them a preferable choice in various sensing applications.

## Figures and Tables

**Figure 1 nanomaterials-14-01534-f001:**
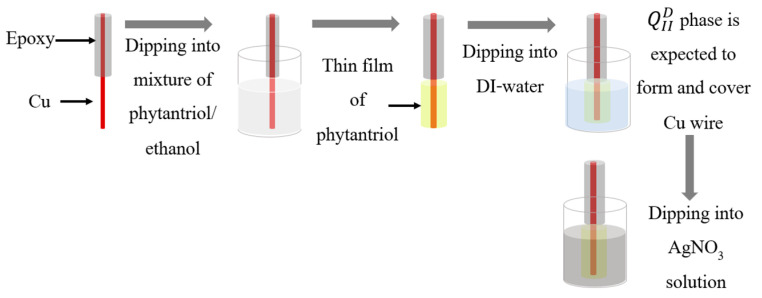
Synthesis steps of AgNDs through phytantriol template.

**Figure 2 nanomaterials-14-01534-f002:**
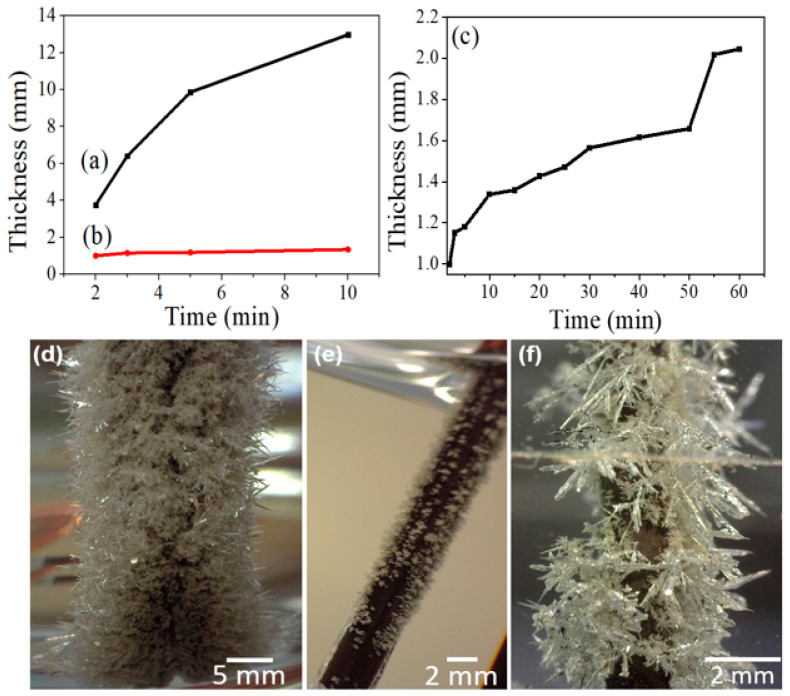
Thickness vs. time for (**a**) AgNDs and (**b**) P/AgNDs for 10 min and (**c**) P/AgNDs for 60 min, whereas (**d**–**f**) corresponds to the optical micrographs of (**a**–**c**), respectively.

**Figure 3 nanomaterials-14-01534-f003:**
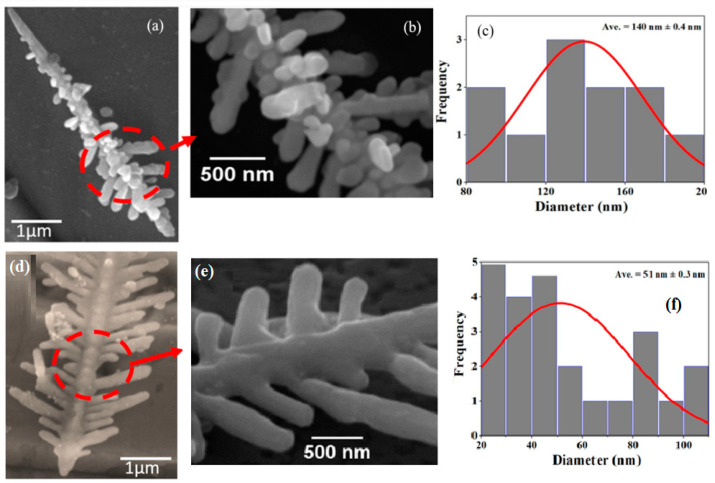
SEM images of (**a**) AgNDs, (**b**) zoomed view of AgNDs, (**c**) particle size distribution of (**a**) and (**d**) P/AgNDs, (**e**) zoomed view of P/AgNDs (**f**), and particle size distribution of (**d**). Curves are the Gaussian fits.

**Figure 4 nanomaterials-14-01534-f004:**
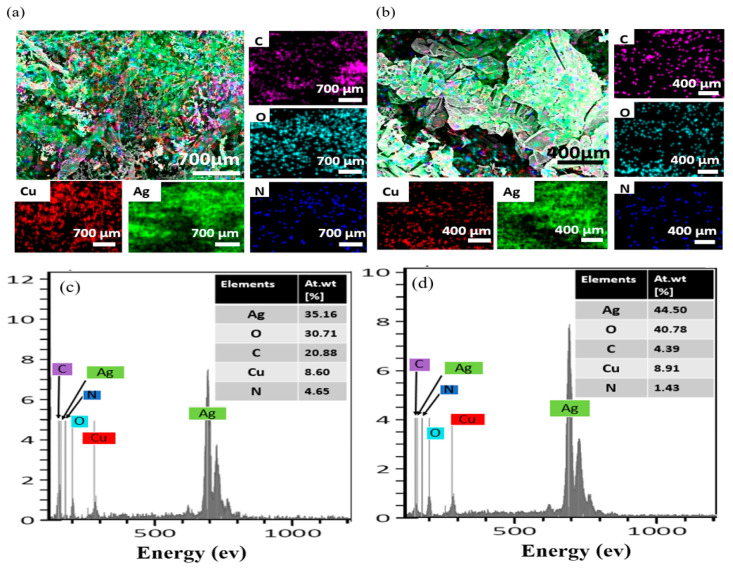
EDX mapping of (**a**) AgNDs and (**b**) P/AgNDs (**c**,**d**) corresponds to spectra of AgNDs and P/AgNDs, respectively. Insets relate to the atomic weight (%) of elements.

**Figure 5 nanomaterials-14-01534-f005:**
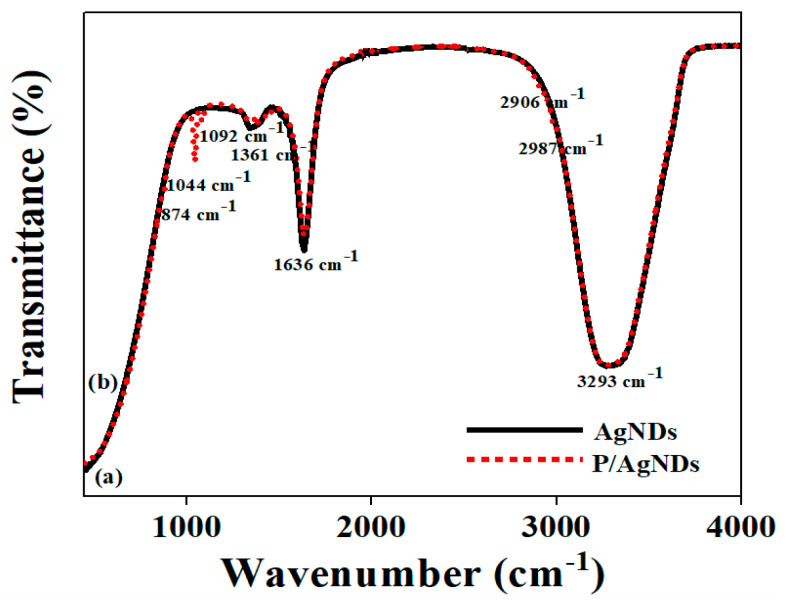
FTIR bands of (**a**) AgNDs and (**b**) P/AgNDs.

**Figure 6 nanomaterials-14-01534-f006:**
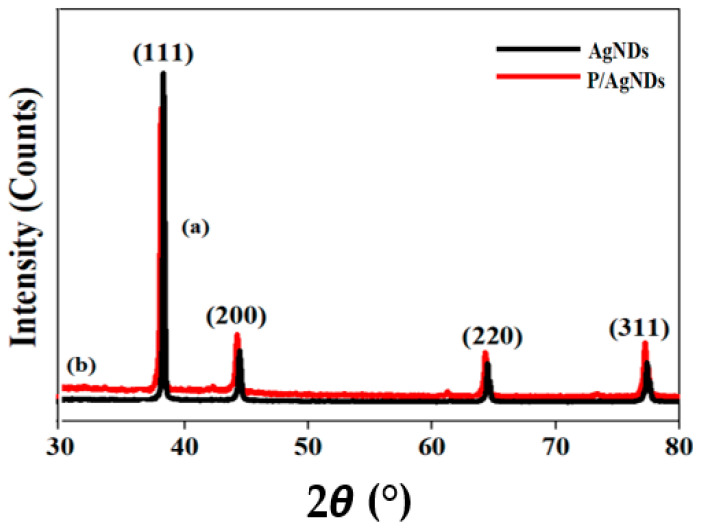
XRD pattern of (**a**) AgNDs and (**b**) P/AgNDs.

**Figure 7 nanomaterials-14-01534-f007:**
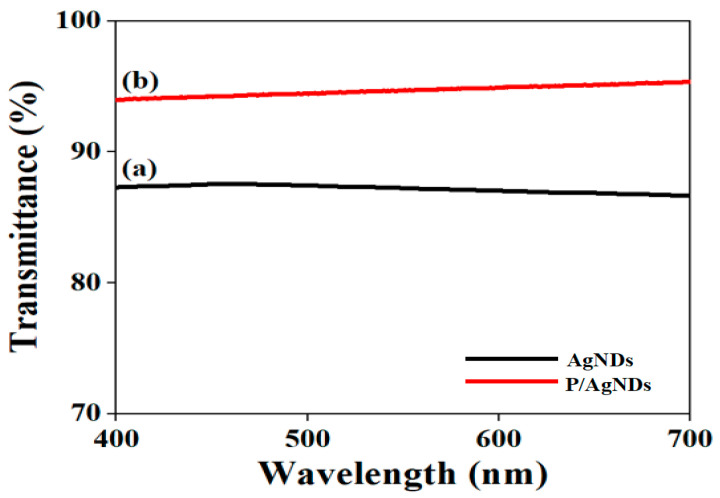
Transmittance of (**a**) AgNDs and (**b**) P/AgNDs.

**Figure 8 nanomaterials-14-01534-f008:**
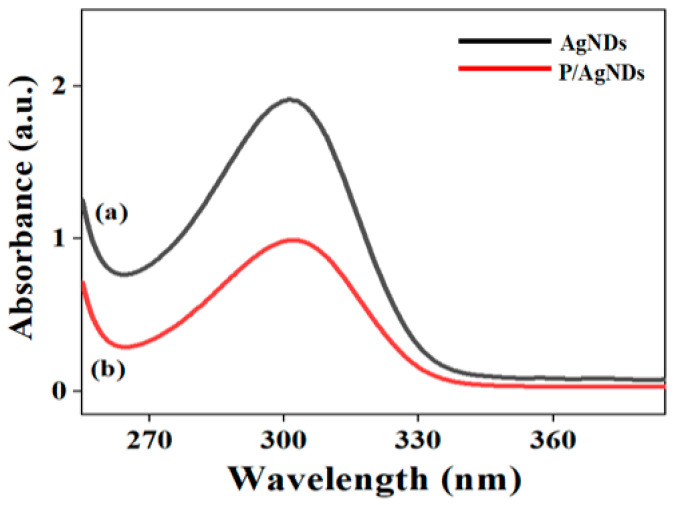
Absorption of (**a**) AgNDs and (**b**) P/AgNDs.

## Data Availability

Dataset available on request from the authors.

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
