# Peer review of "In Situ Rapid Hierarchical Growth of Ag Nanodendrites in Phytantriol: Influence on Polydispersity and Optical Characteristics"

_nanomaterials, 2024, doi:10.3390/nano14181534_

Round 1

Reviewer 1 Report

Comments and Suggestions for Authors

This paper presents a novel room-temperature synthesis of silver nanodendrites (AgNDs) using a phytantriol template, eliminating the need for electrodeposition or high-temperature conditions. The study compares AgNDs synthesized with and without the template, revealing that phytantriol-templated AgNDs (P/AgNDs) exhibit more uniform and well-ordered growth with smaller particle sizes. The synthesis process involves coating a Cu wire with a phytantriol-ethanol mixture, forming a cubic mesophase template in water, followed by the introduction of silver nitrate (AgNO₃) to form AgNDs. The resulting P/AgNDs show high transmittance (up to 95% at 550 nm) and uniform elemental distribution, indicating their potential for various applications. This method offers a simple and effective approach to producing high-quality nanomaterials with broad technological applicability. However, several key points need to be addressed before this manuscript can be considered for publication.

1. The manuscript discusses the hierarchical growth of AgNDs and P/AgNDs, attributing the formation of smaller, well-ordered particles in the presence of a phytantriol template to diffusion-limited aggregation. However, the precise mechanism governing this hierarchical growth, especially the role of phytantriol in reducing the size of nanoparticles, remains inadequately explained. A more detailed discussion on the kinetic aspects of the growth process and how the phytantriol template influences these kinetics would be valuable.

2. The manuscript highlights the superior transmittance of P/AgNDs compared to AgNDs. However, the optical characterization lacks an analysis of how the structural differences between AgNDs and P/AgNDs contribute to their distinct optical behaviors. The influence of nanoparticle size distribution and shape on the observed transmittance and absorption spectra should be more rigorously analyzed.

3. In Figure 3, the red line in the histograms seems to represent a fit to the particle size data, but the method used to obtain this line is not explained in the manuscript. Was it a simple Gaussian fit, or another method? Additionally, the particle size distribution in (b) appears uniform, while (f) shows two peaks around 40 nm and 80 nm. These features are not discussed in the text. Could you please explain the origin of these peaks and the fitting method used?

4. The EDX mapping and FTIR spectra suggest the formation of chemical bonds between the phytantriol and AgNDs. However, the manuscript does not fully explore the implications of these interactions. For instance, how do these chemical bonds influence the stability or potential applications of the synthesized P/AgNDs? Further discussion on the impact of these interactions on the material properties would be beneficial

Comments on the Quality of English Language

The quality of English in the manuscript is generally understandable, but there are areas where clarity can be improved. Some sentences are overly complex or unclear. I recommend moderate editing to simplify the language, improve sentence structure, and ensure that technical terms and concepts are presented clearly. With some revision, the language quality can meet the standards required for publication.

Author Response

  1. The manuscript discusses the hierarchical growth of AgNDs and P/AgNDs, attributing the formation of smaller, well-ordered particles in the presence of a phytantriol template to diffusion-limited aggregation. However, the precise mechanism governing this hierarchical growth, especially the role of phytantriol in reducing the size of nanoparticles, remains inadequately explained. A more detailed discussion on the kinetic aspects of the growth process and how the phytantriol template influences these kinetics would be valuable.

 Ans: Respected reviewer, the manuscript (section results and discussion) has been modified and highlighted according to your suggestions.

  1. The manuscript highlights the superior transmittance of P/AgNDs compared to AgNDs. However, the optical characterization lacks an analysis of how the structural differences between AgNDs and P/AgNDs contribute to their distinct optical behaviors. The influence of nanoparticle size distribution and shape on the observed transmittance and absorption spectra should be more rigorously analyzed.

Ans: The discussion has been addressed in the manuscript accordingly.

Furthermore, the shape and size distribution of nanoparticles also influenced the absorption and transmittance of the material due to the interaction of light with the nanoparticles. Oh et al [31] reported that subwavelength-sized particles can form a homogeneous layer with an effective refractive index between that of Ag and air. This layer could smooth the transition in refractive index from air to the substrate, thereby reducing surface reflectance and consequently increasing transmittance. Therefore, P/AgNDs exhibited high transmittance due to their small size compared to AgNDs

The absorption spectra of AgNDs were largely governed by their plasmonic characteristics, which influenced by the shape, size, and morphology of the dendritic structures. For AgNDs, absorption peak position can vary based on the size and density of the dendrite arms. Smaller branches tend to shift the LSPR peaks toward the blue region, while larger branches shift the peaks toward the red region. Aggregation can also amplify the absorption intensity due to enhanced surface area and plasmon coupling. Figure 8(a, b) shows the abortion spectra of AgNDs and P/AgNDs, respectively. Both samples illustrated a strong SPR band with good symmetry at wavelengths 301 nm and 303 nm, suggesting small, spherical, and narrow-size distribution interval-based nanoparticles. Mukherji et al. [32] reported the Ag nano prisms by the absorption peak at 397 nm. Agnihotri et al. [33] reported SPR peaks at 393 nm, 394 nm, and 398 nm corresponding to AgNPs of average size 5 nm, 7 nm, and 10 nm, respectively. Furthermore, a change in the refractive index around the AgNDs can shift to the peak position, making them useful in sensing applications.

  1. In Figure 3, the red line in the histograms seems to represent a fit to the particle size data, but the method used to obtain this line is not explained in the manuscript. Was it a simple Gaussian fit, or another method? Additionally, the particle size distribution in (b) appears uniform, while (f) shows two peaks around 40 nm and 80 nm. These features are not discussed in the text. Could you please explain the origin of these peaks and the fitting method used?

 Ans: Respected Reviewer, the dendritic structure consists of particles with varying sizes. Therefore, to represent the overall size distribution, we use the Gaussian particle distribution curve to calculate the average particle size. This allows for a more accurate depiction of the particle size distribution in the sample. Moreover, the discussion has been added in the manuscript related to distribution.

  1. The EDX mapping and FTIR spectra suggest the formation of chemical bonds between the phytantriol and AgNDs. However, the manuscript does not fully explore the implications of these interactions. For instance, how do these chemical bonds influence the stability or potential applications of the synthesized P/AgNDs? Further discussion on the impact of these interactions on the material properties would be beneficial

Ans: The discussion has been added in the EDX and FTIR sections accordingly

The quality of English in the manuscript is generally understandable, but there are areas where clarity can be improved. Some sentences are overly complex or unclear. I recommend moderate editing to simplify the language, improve sentence structure, and ensure that technical terms and concepts are presented clearly. With some revision, the language quality can meet the standards required for publication.

Ans: The manuscript has been modified

Reviewer 2 Report

Comments and Suggestions for Authors

This study explored the use of phytantriol as a template to control the synthesis of silver nanodendrites (AgNDs) at room temperature. The presence of phytantriol, due to its water channel effect, resulted in smaller and more uniformly distributed P/AgNDs compared to those synthesized without the phytantriol template. The AgNDs were characterized using TEM, SEM, EDS-mapping, ATR-FTIR, and UV-Vis spectroscopy. However, the study primarily reported the results without providing a detailed explanation of the underlying synthesis mechanism of the AgNDs or how the water channels inhibited the growth of AgNPs. The oxidation state of the synthesized P/AgNDs was also not discussed. Moreover, the manuscript lacked an exploration of practical applications, which is crucial for evaluating the advantages of this synthesis method compared to other AgNDs. Consequently, this paper is not recommended for publication in nanomaterials.

Author Response

Comments: This study explored the use of phytantriol as a template to control the synthesis of silver nanodendrites (AgNDs) at room temperature. The presence of phytantriol, due to its water channel effect, resulted in smaller and more uniformly distributed P/AgNDs compared to those synthesized without the phytantriol template. The AgNDs were characterized using TEM, SEM, EDS-mapping, ATR-FTIR, and UV-Vis spectroscopy. However, the study primarily reported the results without providing a detailed explanation of the underlying synthesis mechanism of the AgNDs or how the water channels inhibited the growth of AgNPs. The oxidation state of the synthesized P/AgNDs was also not discussed. Moreover, the manuscript lacked an exploration of practical applications, which is crucial for evaluating the advantages of this synthesis method compared to other AgNDs. Consequently, this paper is not recommended for publication in nanomaterials.

Ans: The details of mechanism, oxidation, more characterization and advantages of this method has been added in the manuscript accordingly

Reviewer 3 Report

Comments and Suggestions for Authors

Kindly see the attached files.

Author Response

Herein, the author successfully synthesizes silver nano dendrites P(AgNDs) at room temperature without electrodeposition. These empirical results suggest that hierarchical P/AgNDs based on room temperature have significant promise for various applications.

Introduction

  1. What is the novelty, and why it is essential to carry out this research is Include it in the last paragraph of the introduction section.

Ans: It is highlighted in the last paragraph of the introduction accordingly

In contrast to the mentioned reports, reported reaction procedure does not require any supplementary reagents like reductants, or surfactants, or involve stringent conditions such as high temperatures or light exposure. Hence, it is of significance and interest to propose an innovative approach that fulfills the assumption without these requirements.

  1. Rewrite the Introduction section, focusing on and highlighting the importance of the material, technique, and methods acquired and mentioning the positive aspects of carrying out this

Ans: In the current study, the oxidation-reduction reaction with a phytantriol template has been used to synthesize AgNDs hierarchies on Cu wire. Further discussion has been added in the manuscript according to your guidance

  1. Besides this, kindly also mention examples from previous literature reviews. Cite and review the following article to understand how to write an introduction. The following points must be considered while writing an

"Morphological influence of TiO2 nanostructures (nanozigzag, nanohelics and nanorod) on photocatalytic degradation of organic dyes; Volume 400, 1 April 2017, Pages 184-193"

  • General discussion about the topic
  • Previous literature examples
  • material importance with examples
  • Technique/method importance
  • Novelty of your work

Ans: The reference has been cited accordingly.

Experiment:

  1. The authors mentioned the epoxy but did not mention the name. Kindly clearly mention what kind of epoxy is

Ans: Epoxy Resin-Graffiti Resin Company, USA

Results and Discussion

  1. Figure 1: Kindly provide an enlarged optical analysis of AgNDs and P/AgNDs to visualize the thickness variation easily over

Ans: The manuscript has been modified

  1. How do authors calculate mean diameter in Figures 3(c) and 3(f) using software and plot the particle size analysis?

Ans:  These nanoparticles were calculated by the software ImageJ and then plotted by origin.

  1. Provide XRD of AgNDs and P/AgNDs

Ans: The XRD analysis and discussion has been added accordingly

  1. What does a and b represent in Figure 5? Label the

Ans: The manuscript has been modified

Units are missing in Figure 5. Kindly mention them. i,e., cm-1

Ans: The manuscript has been modified

Why is the transmittance higher in the case of P/AgNDs than in AgNDs?

Ans: The transmittance is higher due to porous structure of P/AgNDs, highlighted in the manuscript.  Moreover, discussion has been added in the manuscript accordingly.

Round 2

Reviewer 2 Report

Comments and Suggestions for Authors

While the author offers a solid discussion on the growth mechanism and the benefits of the proposed materials, the potential applications are not addressed in sufficient detail. Figure 8 demonstrates that the prepared samples exhibit distinct SPR bands, which could be utilized as sensing probes. However, whether the Y-axis represents absorption or absorbance is unclear, which creates confusion. Additionally, since any AgNPs with SPR bands can function as detection agents through aggregation, it remains unclear why the proposed P/AgNPs should be preferred. Therefore, I recommend including an application section in the manuscript's conclusion to justify the selection of P/AgNPs.

Author Response

While the author offers a solid discussion on the growth mechanism and the benefits of the proposed materials, the potential applications are not addressed in sufficient detail. Figure 8 demonstrates that the prepared samples exhibit distinct SPR bands, which could be utilized as sensing probes. However, whether the Y-axis represents absorption or absorbance is unclear, which creates confusion.

Ans: Respected reviewer, thank you to raise this point, introduction, results and discussion, and conclusion has been modified and highlighted in the manuscript according to your guidance and suggestions.

Reviewer 3 Report

Comments and Suggestions for Authors

The authors have revised the manuscript; however, a few minor issues should be addressed before its final acceptance.

1.     Page 3, line 104 Rewrite the statement; it is incomplete.

Phytantriol was used as the template, so no reaction between Ag and phytantriol occurred as expected.

2.     In Figure 2 and Figure 4, "bottom left " and Top left, there is some line or presence of words. Kindly delete it.

3.     In Figure 5, The bottom line should be drawn as a solid line (complete), indicating it is the main or reference curve. The top line should be represented as a dotted line, suggesting that it mimics or follows the same pattern but is not the primary focus. This kind of differentiation helps to distinguish between the curves visually.

4.     Page 7, line number 280 is in the wrong place

5.     In Figure 6, kindly utilize the symbol '°'instead of writing out the word 'degrees' in full."

Author Response

The authors have revised the manuscript; however, a few minor issues should be addressed before its final acceptance.

  1. Page 3, line 104 Rewrite the statement; it is incomplete.

Phytantriol was used as the template, so no reaction between Ag and phytantriol occurred as expected.

Ans: The changes have been made in the manuscript accordingly.

  1. In Figure 2 and Figure 4, "bottom left " and Top left, there is some line or presence of words. Kindly delete it.

Ans: The changes have been made in the manuscript accordingly. This is from the journal template.

  1. In Figure 5, The bottom lineshould be drawn as a solid line (complete), indicating it is the main or reference curve. The top line should be represented as a dotted line, suggesting that it mimics or follows the same pattern but is not the primary focus. This kind of differentiation helps to distinguish between the curves visually.

The manuscript has been modified according to your guidance.

  1. Page 7, line number 280 is in the wrong place

Ans: The changes have been made in the manuscript accordingly. This is from the journal template.

  1. In Figure 6, kindly utilize the symbol '°'instead of writing out the word 'degrees' in full."

Ans: The manuscript has been modified according to your guidance.

Round 3

Reviewer 2 Report

Comments and Suggestions for Authors

Accept